# Automatic Registration Algorithm for the Point Clouds Based on the Optimized RANSAC and IWOA Algorithms for Robotic Manufacturing

**Guanglei Li** [1,2,*], **Yahui Cui** [1], **Lihua Wang** [1] **and Lei Meng** [1]

1   School of Mechanical and Precision Instrument Engineering, Xi'an University of Technology,
    Xi'an 710048, China
2   School of Mechanical Engineering, Nanjing Vocational University of Industry Technology,
    Nanjing 210023, China
*   Correspondence: leeglei@gmail.com

**Abstract:** In order to solve the problems of low accuracy and low efficiency of point cloud registration for stereo camera systems, we propose a binocular stereo camera point cloud registration method based on IWOA and Improved ICP. We propose the following approaches in this paper—the registration process is divided into two steps to complete the initial coarse registration and the exact registration. In the initial registration stage, an improved Whale Optimization Algorithm (IWOA) based on nonlinear convergence factor and adaptive weight coefficients was proposed to realize the initial registration in combination with the RANSAC algorithm, and the obtained transformation matrix was used as the initial estimate of the subsequent exact registration algorithm. In the second step of the exact registration stage, an IICP algorithm with the introduction of normal vector weighting constraints at key points was proposed for achieving point cloud exact registration. This algorithm was verified by using Stanford point clouds (bunnies and monkeys) and our own point clouds algorithm, and the proposed algorithm in this paper has high registration accuracy, improved registration speed, and convergence speed.

**Keywords:** IWOA; improved; RANSAC; ICP; point cloud registration

## 1. Introduction

Point cloud registration, widely used in various fields including industrial vision, reverse engineering, cultural relics restoration, virtual reality, intelligent processing, flexible manufacturing, etc., is a key technology in the fields of binocular vision inspection and intelligent processing [1,2]. The purpose of point cloud alignment [3] is to unify multiple point clouds of the same object or scene at different times, different positions, and different angles into the same reference coordinate system by finding some spatial transformation relationship, to realize the fusion and stitching between point clouds and obtain a complete 3D point cloud of the target object. Considerable relevant research has been carried out. Hafiz et al. [4] proposed a novel method for the conversion of 2D RGB images to 3D point clouds by constructing a single-code-multiple-decode depth network architecture in which each decoder generates certain fixed viewpoints and, subsequently, fuses all the viewpoints to generate a dense point cloud, which was experimentally shown to have superior performance. The iterative closest point (ICP) algorithm proposed by Besl [5] et al. in 1992 was extensively applied to point cloud matching, with the focus placed on searching for the correspondence between two sets of registration points for calculating the transformation matrix. However, the ICP algorithm is subject to the disadvantages of necessary inclusion relationships, low computational efficiency, and high initial location requirements [6]. Moreover, the registration results of the algorithm depend heavily on the initial position of the point cloud. When the position of the two-point cloud data

sets differs greatly, the results may converge to the optimal local solution and even lead to failure in registration. Therefore, a good initial value is required for the algorithm to ensure convergence.

In order to address the above-mentioned problems, scholars have made improvements based on the ICP algorithm, and brought forth lots of new point cloud registration methods. Yu Wenli et al. [7] accomplished the initial registration of point cloud using the four PCS method, and calculated the rigid transformation matrix of the three-dimensional model using the point-to-face ICP method and least square method, achieving some improvements; Wang Yujian et al. [8] proposed a registration method based on the multi-layer index structure of octree and KD tree and improved the efficiency and accuracy of point cloud registration. Wang Dong et al. [9] reduced the matching error at the edge and corner points and improved the overall matching effect by combining the ICP algorithm and inverse method of surface point parameters. Segal et al. [10] integrated the point-to-point and point-to-surface ICP algorithms into a statistical framework and greatly improved the robustness of three-dimensional point cloud registration for noise, outliers and mismatching; Yang et al. [11] adopted a nested branch and bound search structure comprising two branch and bound searches, one of which was the Go-ICP algorithm bounded by an external branch for rotation space search, with the highest accuracy but still exposed to the inferior strength of time-consuming in the case of calculating a large number of point clouds.

With the algorithm mentioned above, some progress has been made in the study of point cloud registration, but the real-time and accuracy of the point cloud are increasingly demanding as intelligent manufacturing moves forward. Problems faced by point cloud registration, including poor real-time registration, low accuracy, and poor robustness, are still to be solved, thus making the research on rapid and accurate registration methods a hot spot and a difficult point in the engineering field. The problems of determining the corresponding points in the coarse alignment stage and solving the translation and rotation matrix in the fine alignment stage can be converted into constrained optimization problems in the final analysis. Conventional solutions are subject to problems such as failing to find optimal solutions for constraints, being prone to local optimization and poor real-time capability, etc. The application of an intelligent optimization algorithm has been a novel idea in the study of point cloud registration in recent years. Thus far, relevant research has been conducted, and corresponding progress has also been made. Yang Bo et al. [12] put forward an improved ICP point cloud registration algorithm based on the genetic algorithm, which is provided the ability of optimal global search for rough registration, registers point cloud data with the 3D model, and provides a good iterative initial value for subsequent accurate registration. Experimental results showed that this method could effectively improve the accuracy and rate of point cloud registration. Huang et al. [13] fulfilled point cloud registration using an iterative closest point algorithm based on hierarchical particle swarm optimization to improve the performance of 3D point cloud registration, then obtained the feature points precisely expressing the point cloud structure by jointly searching for the optimal particles using the curvature of the point cloud as the fitness value, and finally, performed the registration of the feature points using the iterative closest point algorithm and achieved better registration efficiency. Based on the ICP algorithm, Wu Hao [14] proposed an automatic registration algorithm that carries out automatic rough registration in view of boundary characteristics of the point cloud, followed by automatic fine registration through an improved ICP algorithm. Experimental results proved that this method possesses a sub-millimeter improvement versus the classical algorithm.

The ICP algorithm presupposes that the coarse registration process establishes the correct initial registration point pairs and achieves accurate registration of point clouds. The quality of rough registration directly affects the efficiency and accuracy of fine registration. Until now, the coarse registration methods mainly include Gaussian-likelihood estimation factor analysis proposed by Li Zan et al. [15], 3D-NDT-based point cloud registration by Zhang Xiao et al. [16], the feature-based method by Tian et al. [17], and RANSAC algorithm by RUSU et al. [18]. Liu Meiju et al. [19] combined the descriptor algorithm for

internal morphology with the histogram algorithm with fast point characteristics and then improved the RANSAC algorithm for the initial registration of point cloud by means of pre-estimation and 3D grid segmentation to obtain better results. Ren et al. [20] proposed a color-based point cloud alignment algorithm that extracts the hue components based on the color information of the point cloud and makes the hue distribution of the tangent plane continuous. The error function consists of the color of the point cloud to be aligned and the geometric error. The error function is optimized using the Gauss–Newton method, and the algorithm has good robustness under different lighting conditions. Choi et al. [21] proposed a new algorithm to improve numerical stability. Unlike previous algorithms that rely heavily on point-to-plane distances, their algorithm constructs a cost function based on two different projection distances, and experiments show that the algorithm has high accuracy for color point cloud alignment. Ge et al. [22] proposed a pairwise non-rigid alignment algorithm for 3D point clouds, which generates a correspondence between two deformed point clouds based on the isometric deformation property invariant, and subsequently uses a random sample consensus (RANSAC) algorithm for cloud point feature extraction to better prepare for the optimization of alignment. Yang et al. [23] proposed a spatially decomposed and optimized RANSAC algorithm for indoor plane detection based on the phenomenon that the traditional RANSAC algorithm is prone to feature extraction errors for occluded and missing point cloud data. The method uses a weighted PCA method to estimate the normal vector of the point cloud and then employs angular clustering to partition the indoor space, followed by an optimized RANSAC method to calculate the alignment parameters from the obtained point cloud data. Ghahreman et al. [24] proposed a robust method for direct analysis of point cloud data, which proposes an improved RANSAC algorithm to overcome the adverse effects of point cloud data, such as loss and occlusion. The plant feature extraction experiment proved the effectiveness of this algorithm. Niloy et al. [25] investigated the effects of neighborhood size, curvature, sampling density, and noise on normal estimation and proposed an initial position selection based on the curvature of the point cloud surface, which can effectively improve the robustness of the ICP alignment algorithm. Takeshi et al. [26] proposed a new point cloud alignment and data segmentation algorithm with random sampling and the Least Median of Squares (LMS) or the least-median-of-squares (LMedS) estimator. The algorithm was robust in the presence of outliers (outliers) such as noise and occlusion. Rantoson et al. [27] improved the point cloud alignment by considering a new discrete curvature parameter, which also divides the alignment process into two steps: coarse alignment and exact alignment, for which the coarse alignment is performed based on the enhanced Hough transform (HT) and the improved RANSAC model transformation. In the exact alignment stage, the author proposed a new variant of the ICP method to reduce the alignment error when using the curvature parameter. The algorithm they used took into account the curvature similarity and the Euclidean distance to define the criteria used to search for correspondences. Donoso et al. [28] from Australia compared common improved ICP-based algorithms to scan the scene at 20 Hz using their own Velodyne HDL-64E scanner installed on a minecart, and none of the improved ICP-based algorithms could achieve alignment accurately, precisely, and quickly at the same time in three evaluation metrics based on accuracy, precision, and relative computational cost. The best-performing improvement algorithm employs a strategy of filtering the dataset, uses normal forms of local surface geometry, uses the distance between points in a point cloud and the corresponding surface in a reference point cloud as a measure of the fit between the two point clouds, and points out the limitations of various existing improvement algorithms based solely on ICP for terrain mapping. The above-mentioned algorithms have their characteristics for feature extraction, but there are fewer point cloud features that have been applied to processing. There are still many research and studies that are needed to address some of the challenges and difficulties in pre-processing and alignment of point cloud data, considering the time-sensitiveness and accuracy of feature identification in processing.

In the field of robot vision processing, the vision system extracts the point cloud data of the target workpiece in real-time, identifies the features of the workpiece through the calculation of the point cloud data, and then determines the area to be processed and the reserved area. The ICP algorithm alignment accuracy is high, but the algorithm requires the point cloud to be aligned to have a certain initial relationship, and the ICP alignment algorithm alone is easy to fall into misalignment, and the alignment is not high in real time. In the field of robot processing, the surface characteristics of the workpiece and the model data distribution can be described by some model parameters, so the RANSAC algorithm is faster, but it is an uncertain algorithm that has a certain probability of yielding a reasonable result; therefore, the RANSAC algorithm can be used as the basis of the initial alignment algorithm. Based on the above, a new hybrid point cloud registration algorithm integrating the whale optimization RANSAC algorithm and ICP algorithm for point cloud initial alignment was hereby presented in this paper. First, the optimal solution of the RANSAC algorithm is solved quickly by the improved whale optimization algorithm, and the coordinate translation and rotation parameters are obtained by the improved ICP algorithm. Simulations and experiments verified the registration accuracy and efficiency of the algorithm designed in this paper.

The rest of this paper consists of four sections. Section 2 introduces the RANSAC algorithm to achieve initial alignment and proposes a nonlinear convergence factor and adaptive weight coefficient method to improve the whale optimization algorithm to optimize the RANSAC algorithm in order to achieve initial alignment quickly and accurately. Section 3 is the point cloud accurate alignment and proposes a method to improve the ICP algorithm based on the key point normal vector weighted judgment method. Section 4 describes our experiments by comparing the RANSAC algorithm, ICP algorithm, Depth Filtering-ICP, and IWOA-RANSAC-ICP algorithms, and it was concluded that the proposed method in this paper has a small mean square error, low false identification rate, and faster alignment speed. The effectiveness and superiority of the algorithms in this paper are proved. Section 5 summarizes the research in this paper and suggests future research directions.

## 2. Optimized RANSAC Algorithm Using IWOA

### 2.1. RANSAC Algorithm

The requirement for a typical ICP algorithm to register the initial position of the point cloud is rather demanding, and it is easy to fall into local optimization when there is a large initial position deviation. In this study, the RANSAC algorithm was used for coarse registration in the initial registration stage. This very algorithm aims to find an optimal transformation matrix for eliminating the error matching and solve 12 unknown parameters of the transformation matrix by randomly selecting six matching points using the normalization method. The general expression of the transformation matrix *T* is:

$$\begin{bmatrix} x' \\ y' \\ z' \\ 1 \end{bmatrix} = \begin{bmatrix} R_{11} & R_{12} & R_{13} & T_x \\ R_{21} & R_{22} & R_{23} & T_y \\ R_{31} & R_{32} & R_{33} & T_z \\ 0 & 0 & 0 & 1 \end{bmatrix} \begin{bmatrix} x \\ y \\ z \\ 1 \end{bmatrix} \tag{1}$$

where $(x', y', z')$ and $(x, y, z)$ are the cartesian coordinates of the target point cloud and the point cloud to be registered.

Given a large number of point clouds, the existence of many mismatching points, and the poor timeliness, coupled with the fact that the solution obtained within the specified iterations may not be optimal, the accuracy of the initial registration is affected while solving the transformation matrix by the RANSAC algorithm.

### 2.2. Improve WOA Based on Nonlinear Convergence Factor

Whale optimization algorithm (WOA) is a new swarm intelligence algorithm proposed by Australian scholars Seyedal et al. [29] based on the predatory behavior of humpback

whales, which, compared with most optimization algorithms, is characterized by strong global optimization capability and simple structure. Inspired by bionics, the algorithm models the featured hunting methods of whales as the processes of encirclement, predation, and random search. The basic model of this algorithm is described as follows: Supposing that the number of whale individuals in d-dimensional search space is *N*, the *i* th individual in the *t* th iteration is represented as:

$$X_i^t = (x_i^1, x_i^2, \cdots)(i = 1, 2, \cdots, m; t = 1, 2, \cdots, T_{max}) \tag{2}$$

where $T_{max}$ is the maximum times of the iterations.

The encirclement and contraction phase simulates the behavior of humpback whale populations identifying the prey and contracting the encirclement. The behavior is expressed by the formula:

$$D = |C{\cdot}X^*(t) - X(t)| \tag{3}$$

$$X(t+1) = X^*(t) - A{\cdot}D \tag{4}$$

where *A* and *C* denote coefficient vectors; $X^*$ is the current global optimal position; *X* is the position of the individuals in the *t* th generation; and *t* denotes the current number of iterations. Coefficients A and C are calculated as:

$$A = 2a{\cdot}r - a \tag{5}$$

$$C = 2r \tag{6}$$

where *a* represents the accommodation coefficient that decreases linearly from 2 to 0 with the increasing number of iterations; and *r* is the random vector of the [0, 1] distribution. Given that $|A| < 1$ in this phase, the population converges to the current optimal solution.

Whales prey in two ways of spiraling procession and narrowing encirclement. Assume that the probability of both methods is fifty-fifty when they carry out a task, the position can be updated as:

$$X(t+1) = D'{\cdot}e^l{\cdot}\cos(2\pi l) + X^*(t) \tag{7}$$

$$D' = |X^*(t) - X(t)| \tag{8}$$

where $D'$ denotes the distance between the individual and the optimal solution in the *t* th Generation; and *l* is a random number from [–1, 1].

If $|A| > 1$ while searching for optimal solutions, the position between individuals updates based on each other's positions to avoid falling into the local optimal solution to some extent. The mathematical models of the mechanism are described as follows:

$$D = |C{\cdot}X_{rand}(t) - X(t)|. \tag{9}$$

$$X(t+1) = X_{rand}(t) - A{\cdot}D. \tag{10}$$

where $X_{rand}$ denotes the position vector of random individuals in the population.

It can be concluded from the above algorithms that the global search capability of the conventional WOA is dependent on the convergence factor *a*, whose decreasing tendency with the increasing number of iterations reduces both the convergence rate and the search capability. At the initial stage of the algorithm, a larger *a* endows the algorithm with a greater global search capability. As the algorithm proceeds, a smaller convergence factor *a* can provide a higher search accuracy. Therefore, a nonlinear convergence factor is proposed as:

$$\omega = (1 + \frac{2t}{T_{max}})^5 \tag{11}$$

$$a = (2 - \frac{2t}{T_{max}})(1 - \omega). \tag{12}$$

where $t$ means the current time of iterations; and $T_{max}$, the maximum time of iterations set for the algorithm. The nonlinear convergence factor introduced in this paper produces large parameter $A$ in the early iteration stage, improves the global search capability of the algorithm, and speeds up its convergence; in the latter iteration stage, a smaller parameter $A$ is produced, and the convergence accuracy is improved.

Given that the standard WOA does not take into account the current optimal solution during the iterative process, there may be differences while guiding whales for position updating. WOA combines the idea of population optimization guided by inertia weight in the PSO algorithm, and introduces the adaptive parameters as inertia weight factors in the formula for updating the position, so that the optimal solution can be fully utilized for improving the optimization accuracy of the algorithm. The improved formula for position update is presented as:

$$X(t+1) = \omega \cdot X_{rand}(t) - A \cdot D. \tag{13}$$

where the weighting factor $\omega$ nonlinearly increases with the rising number of iterations, and the algorithm shows greater global search capability in the case of a relatively large weighting factor in the initial stage; the weighting factor is gradually decreasing with the increase in iterations, and in this case, the algorithm carries out a spiral search in the neighborhood of optimal solution using a smaller weight to prevent it from falling into local optimization.

### 2.3. Realize Point Cloud Initial Registration Process

In the realization process of the point cloud coarse registration by RANSAC, an appropriate sample subset can improve the registration efficiency, while the distance error determines the registration accuracy. Six sampling points are selected from $P$, the point cloud to be registered, and the distance between every two points is greater than the preset minimum threshold $d$. In order to obtain a minimal value by the error function, optimal ones should be found from all the transformations to complete the initial registration. In this paper, the improved WOA was used for optimization. However, the problem needs to be first elaborated by a mathematical model—fitness function, i.e., the target of the optimization algorithm. The corresponding point cloud registration error of the algorithm used in this paper and $l$ are defined as:

$$l = \sqrt{\frac{1}{n} \sum_{i=1}^{n} \left[ (x_i^p - x_i^q)^2 + (y_i^p - y_i^q)^2 + (z_i^p - z_i^q)^2 \right]}. \tag{14}$$

The fitness function $f(t)$ is defined as:

$$f(t) = \arg min \sum_{i=1}^{n} H(l) = \begin{cases} 0.5l^2, |l| < m_l \\ 0.5m_l(2|l| - m_l), |l| > m_l \end{cases} \tag{15}$$

where $H(l)$ indicates *Huber* penalty function, $m_l$ denotes the preset threshold, and $l$ is the distance difference after the transformation of the corresponding points in the $i$ th group. The transformation satisfying the minimum value of the error function is considered optimal when the transformation matrix is the initial registration and is used for completing the initial registration of the residual point cloud. The realization flow of $RANSAC$ registration by IWOA is shown in the figure below (Figure 1):

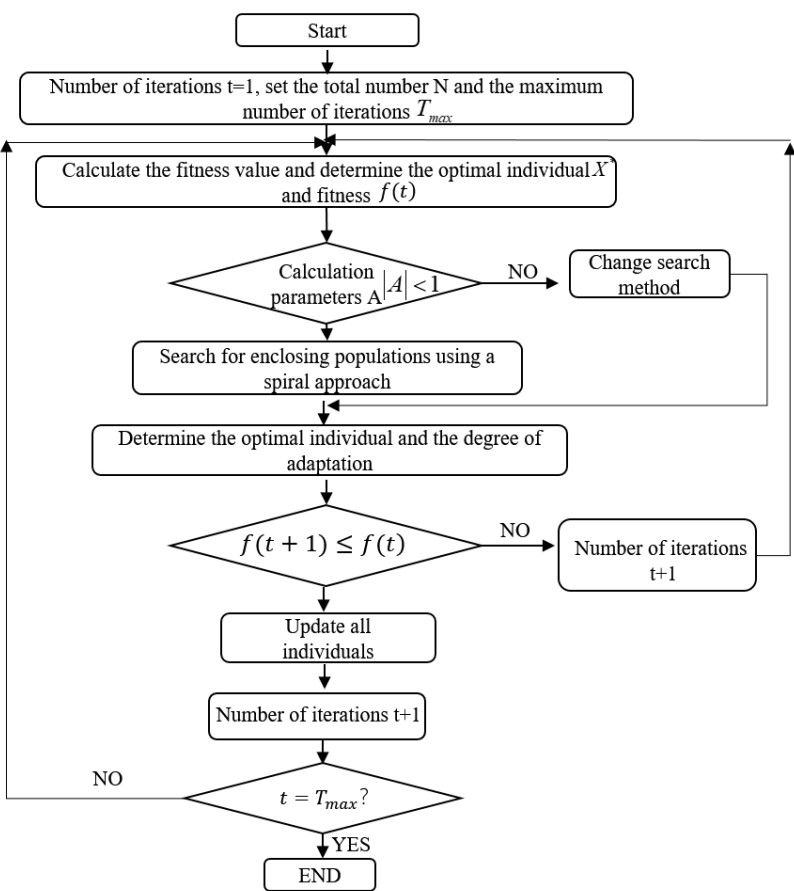

**Figure 1.** Realize Point Cloud Initial Registration Process.

## 3. Realize Point Cloud Fine Registration Using Improved ICP Algorithm

### 3.1. Mathematical Models of the ICP Algorithm

In point-based point cloud registration, the transformation between the target point cloud and the point cloud to be registered can be determined by the affine transformation translation matrix *T* and the rotation matrix *R*. Currently, the iterative closet point (ICP) and various variants of the ICP algorithm are the most widely-used point cloud registration algorithms. The ICP algorithm was first developed by Besl et al. [1], which finds the least-square rigid transformation matrix between two point cloud data sets according to the closest distance criterion of the corresponding point, and repeats the iterations until the local minimum value is obtained. Considerable deformations and optimizations of the ICP algorithm were proposed for different application environments.

The registration of the surface point cloud can be simplified as the matching of measured point cloud relative to the theoretical point cloud. If *P* is set as the measured point cloud of the curved workpiece while *Q* is the theoretical point cloud, the relationship between these two point clouds can be shown as:

$$Q = R \cdot P + T \tag{16}$$

where $R = [R_x, R_y, R_z]^T$ and $T = [T_x, T_y, T_z]^T$ represent the rotation and translation of the three axes (x, y, z) of *P* relative to the theoretical point cloud *Q*, respectively. Select multiple point pairs in the measured point cloud dataset, denoted by $p_i \in P$, and find the corresponding point set $q_i$ in the theoretical point cloud set *Q* so that the value of the target function shown in the following formula can be the minimum:

$$d = min|p_i - q_i| \tag{17}$$

Then, calculate the transformation matrices $R$ and $T$ to minimize value of the error function:

$$f(R,T) = \min_{R,T} \frac{1}{N_P} \sum_{i=1}^{N_P} |q_i - (p_i \cdot R + T)|^2 \tag{18}$$

Rotate and translate the rotation matrix $R$ and translation matrix $T$ obtained in Step 2 for $p_i$ using the transformation matrix for a new corresponding point set:

$$P_i' = \left\{ p_i' = p_i \cdot R + T, p_i \in P \right\} \tag{19}$$

Calculate the average distance between $P_i'$ and the corresponding point set $Q$:

$$d = \frac{1}{N_P} \sum_{i=1}^{N_P} |P_i - Q_i|^2 \tag{20}$$

If the average distance $d$ is less than the preset threshold or the number of iterations is greater than the preset maximum number of iterations $\tau$, leave the loop and stop the iterations; otherwise, return to recalculate the transformation matrix until the convergence requirements are met. The standard flow of ICP registration is shown in Figure 2.

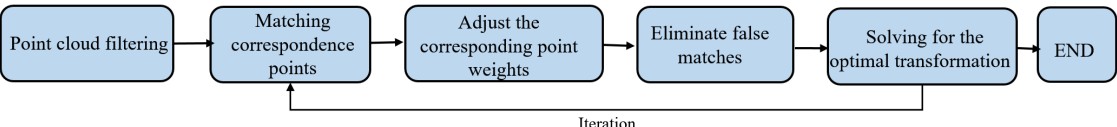

**Figure 2.** Flow of ICP.

The accuracy of the ICP algorithm is generally evaluated as the root-mean-square error $\sigma_{MSE}$ that calculates the distance of all point clouds. The formula is:

$$\sigma_{MSE} = \sqrt{\frac{\sum_{i=1}^{N_P} |q_i - (p_i \cdot R + T)|^2}{N_P}}. \tag{21}$$

The standard ICP algorithm is not necessarily required to segment and extract features from the point cloud data to be processed, which saves time for feature extraction and helps obtain more accurate registration results. Additionally, the algorithm converges well in the presence of better initial registration values. However, in search of corresponding points, a comparison with a large number of point cloud data should be conducted, and large computation is therefore required. It is irrational for the standard ICP to make the hypothesis that the closest point of Euclidean Distance is the corresponding point, which brings about a certain number of corresponding false points [29]. For this reason, an improved ICP algorithm was proposed in this paper.

### 3.2. Realize Accurate Registration of Point Cloud

Conventional point cloud matching algorithms are applicable only between two-point clouds, and the two-point cloud sets have an inclusion relation. In the case of a big difference in the initial spatial position of the overlapping point clouds, the conventional algorithm is apt to fall into the local optimal solution [15,30]. Therefore, under the premise that the standard ICP algorithm satisfies the minimum error function. The algorithms are detailed as follows:

(1) Downsample the point cloud to be registered $P$ and the target point cloud $Q$;
(2) Search for the closest point of the target cloud using KD-tree, calculate the distance $d_i$ in accordance with Equation (17); if $d_i < k$, calculate the normal vector $\theta_i$ of the two corresponding points in $P$ and $Q$, and the point is determined as the corresponding point in $P$ if $\theta_i < \theta$, otherwise, repeat the iteration;

(3)  Calculate the average distance *d* of the corresponding point set. If it is less than the preset threshold, or if the times of iterations are greater than the preset maximum number of iterations $\tau$, leave the loop and stop the iterations; otherwise, return to recalculate the transformation matrix until the convergence requirements are met. The registration flow of the algorithm proposed in this paper is shown in Figure 3.

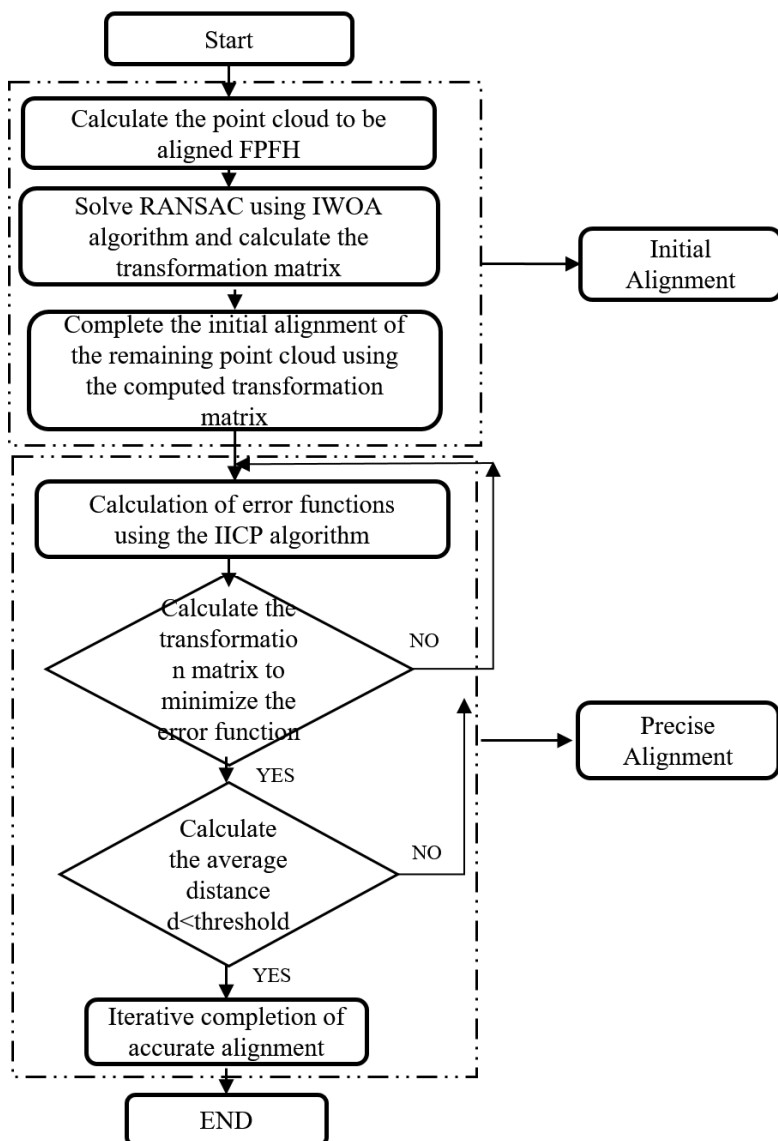

**Figure 3.** Flowchart for the initial registration of RANSAC by IWOA+ accurate registration of the improved ICP.

## 4. Verification and Analysis

### 4.1. Registration Visualization

In order to test the effectiveness of the proposed algorithm, the data sets Bunny (40256) and Monkey (125952) from Stanford University were adopted for simulation. The initial state of the point cloud is shown in Figure 4. The model computer used in the test is I5-10400F, configured with a memory of 16G, a display card of Nvidia GTX 1060 6G, an operating system of win10 64 bit, and is operated on VS2019. The target point cloud was rotated −50 degrees on the Z axis, and shifted 5 mm in X and Y directions, respectively, but −10 mm in the Z direction, which was then used as a point cloud to be registered. The downsampling grid size was set to 0.05 mm. Since the initial alignment results obtained by the method in the literature [21] have a large deviation, the alignment accuracy and

speed were further compared in this paper. and the proposed algorithm was compared to standard ICP, RANSAC algorithms, Depth Filtering-ICP. The initial state of the point cloud is shown in Figure 4.

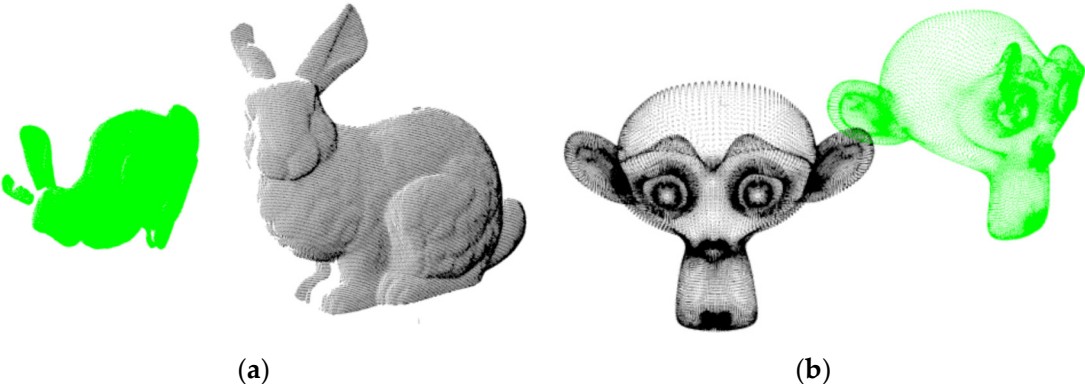

(**a**)　　　　　　　　　　　　　　　　　(**b**)

**Figure 4.** Initial state of point cloud. (**a**) Bunny; (**b**) Monkey.

The maximum number of iterations of ICP, RANSAC, Depth Filtering- ICP and IWOA-RANSAC-ICP algorithms was all set as 50. In IWOA algorithm, set 20 times iterations, 10 whale populations and a 0.05 threshold of fitness function $m_l$ were set. For the research objects of this paper, characteristic quantities $f$ were selected as the evaluation index in order to evaluate the algorithm. It is $[50, 1000]$ for the value range of characteristic quantity $f$ in this paper, which was used for histogram feature quantity evaluating feature points. We can express the feature evaluation function $f_k$ of point cloud as follows:

$$f_k = k_1 + 3 * k_2 \tag{22}$$

Among which $k_1$ is determined by the angle between the normal vector of any point and that of its adjacent point. Moreover, the plane is divided into $[0, \pi/3][\pi/3, 2\pi/3]$ $[2\pi/3, \pi]$, and $k_1$ is noted as 1,2,3 according to the range of angle between the normal vector and that of its adjacent point. $k_2$ is the Euclidean distance between any point and the nearest adjacent point. If the distance is one that is greater than a certain threshold value and zero that is less than the threshold value, the threshold value was set to 0.005 in this paper. Then we established a histogram with an interval number that two multiple by 3 equals six to obtain the corresponding 6-dimensional characteristic value in accordance with the classifications of these two feature values.

We regarded the percentage occupying the total number of the points of the point cloud in each interval as the corresponding interval value and characteristic value. The default values were adopted for other parameters, which $f_k$ were 16 before optimization. Moreover, it was 0.500000 for the fitness function value. It was 0.000015 for the evaluation function value. After optimization, they $f_k$ were 42, 0.239051, and 0.000006, respectively, for the default value, the fitness function value, and the evaluation function value. For convergence curve of the optimized RANSAC algorithm of IWOA, it is shown in Figure 5.

It can be seen from the iteration curve that it can quickly obtain the optimal solution after the start of iteration for the optimized RANSAC algorithm of IWOA. It is less than that of the simple RANSAC algorithm for the fitness value of this algorithm, which has a faster speed of convergence, stable curve, and good real-time performance. The registration results of the four registration methods are shown in Figure 6.

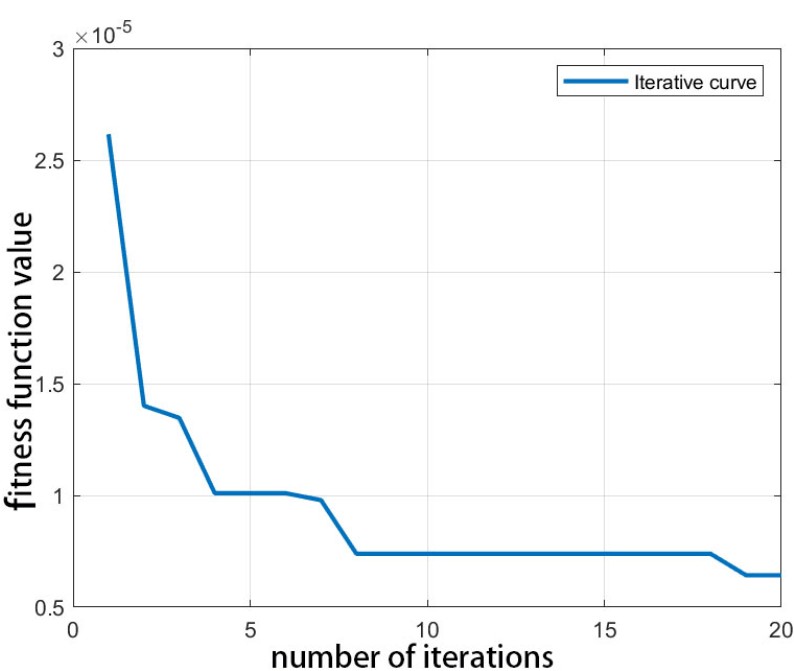

**Figure 5.** Iteration curve of IWOA optimized RANSAC algorithm.

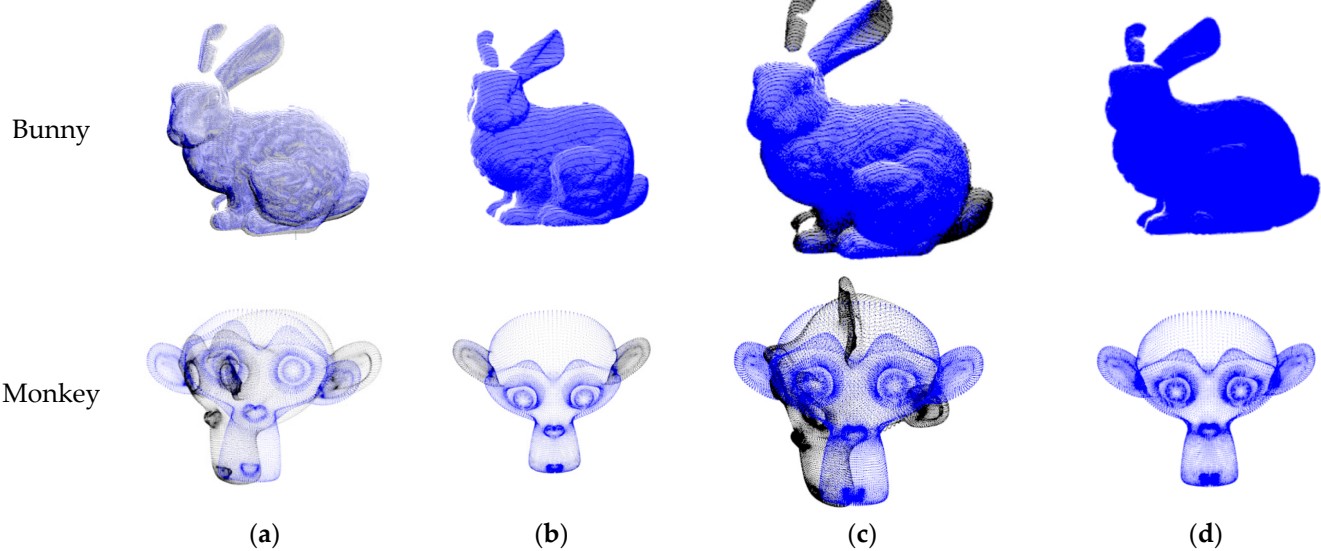

**Figure 6.** Registration results of Bunny and Monkey. (**a**) Standard ICP algorithm; (**b**) Standard RANSAC algorithm; (**c**) Depth Filtering-ICP; (**d**) IWOA-RANSAC-ICP algorithm.

According to Figure 6, the standard ICP algorithm and the three standard algorithms are all capable of completing the registration of Bunny and Monkey point clouds. However, the standard ICP algorithm is greatly influenced by the accuracy of initial registration and poses a great impact on subsequent registration due to the poor accuracy of registration in the process of Monkey registration. In the case of direct registration with ICP, despite the fast rate, there is a large error, and in some cases, it falls into local optimization, failing to fulfill accurate registration. The standard RANSAC algorithm is required to transform and calculate a large amount of point cloud data, which makes its registration rate slower than that of the standard ICP algorithm. However, the proposed IWOA-RANSAC-ICP algorithm overcomes the shortcomings of both the above two algorithms and introduces intelligent optimization algorithms that can quickly achieve the initial registration of point clouds. The point cloud after initial registration subsequently provides a more reliable initial value for the improved ICP algorithm, and the normal vector constraint of key points

is introduced into the ICP registration algorithm to avoid local optimization in the stage of accurate registration. On the whole, this method can achieve the optimum registration effect, but given the long registration process, its efficiency remains to be improved. The registration time of Bunny and Monkey using these four methods is shown in the following table (Table 1).

**Table 1.** Registration time of these three methods (ms).

| Point Cloud | ICP | RANSAC | Depth Filtering-ICP | IWOA-RANSAC-ICP |
|---|---|---|---|---|
| Bunny | 745.52 | 1026.437 | 1652.41 | 1156.532 |
| Monkey | 2044.7 | 2806.24 | 3758.65 | 3025.42 |

As can be seen from the above table, the number of Bunny point clouds is small, and the ICP algorithm converges the objective function by repeated iterations of the points so that the corresponding points are gradually approximated to achieve accurate alignment quickly, and when the number of point clouds increases significantly, the time required by the ICP algorithm increases accordingly. Depth Filtering-ICP and IWOA-RANSAC-ICP take more time to align than the previous two due to the increased workflow of point cloud registration, and in the subsequent research, the focus is on improving and optimizing the algorithm to increase the speed of the alignment. The mean square error of the four methods is shown in Table 2 below.

**Table 2.** Comparison of mean square error between the proposed algorithm and other algorithms.

| Point Cloud | Error | $\sigma_{MSE}$ (mm) | | | |
|---|---|---|---|---|---|
| | | ICP | RANSAC | Depth Filtering-ICP | Algorithm of This Paper |
| Bunny | x | 0.321532 | 0.161132 | 0.752535 | 0.125004 |
| | y | 0.247523 | 0.106212 | 0.532120 | 0.113524 |
| | z | 0.374156 | 0.264652 | 0.332154 | 0.154236 |
| Monkey | x | 2.51821 | 3.152478 | 5.354225 | 2.245156 |
| | y | 0.618039 | 0.872515 | 0.952452 | 0.528965 |
| | z | 1.98336 | 1.502146 | 1.752154 | 0.896534 |

According to Tables 1 and 2, the registration speed of Bunny and Monkey using the standard ICP algorithm is fast but is also subject to a large root-mean-square error, which may be attributed to the great influence of the initial state on the registration. Additionally, the standard RANSAC algorithm is slow but accurate, and the Depth Filtering-ICP algorithm is better than the first two in some metrics. The IWOA-RANSAC-ICP algorithm is close to the standard RANSAC algorithm in terms of the registration rate, but it is more accurate with a minimal root-mean-square error.

### 4.2. Analysis of Experimental Data

In order to test the algorithm's effectiveness in the laboratory scanning point cloud, the data from two groups of mechanical devices were collected using a portable three-dimensional scanner (PRINCE 335) with a reference distance of 300 mm and a field depth of 250 mm. The operating distance of the scanner from the scanned surface ranged from 200 to 450 mm, and the precision reached 0.01 mm + 0.025 mm/m by precise scanning. The number of collected workpiece point clouds was 853,105, and the collecting site and the collection point clouds are shown in Figure 7. The above figure is named Workpiece A, and the below figure, Workpiece B, to simplify the subsequent analysis. They are the same as the above for the algorithm settings.

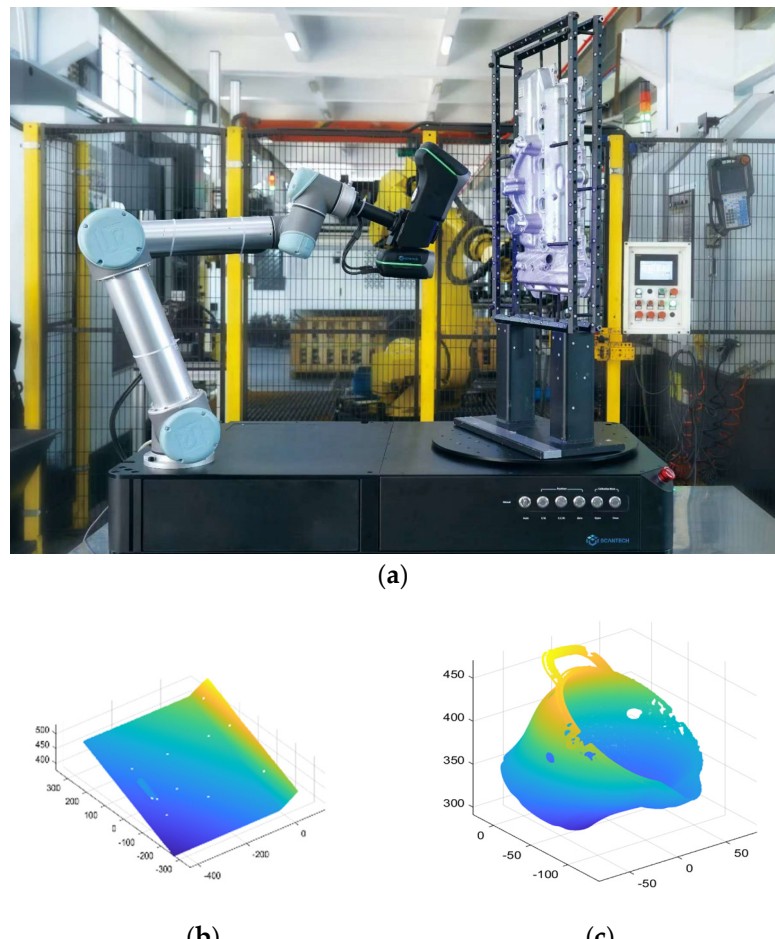

**Figure 7.** Collected surface workpiece point clouds. (**a**) Experimental platform; (**b**) Point cloud data of workpiece A; (**c**) Point cloud data of workpiece B.

Workpiece A has 853,105 points, which is reduced to 648,172 after downsampling. Workpiece B has 356,310 points, which is reduced to 175,640 after the same operation. Two groups of point clouds were processed by downsampling and translation rotation transformation, respectively, and then registered using the three algorithms mentioned above. The results of their registration using the four algorithms are shown in Figure 8.

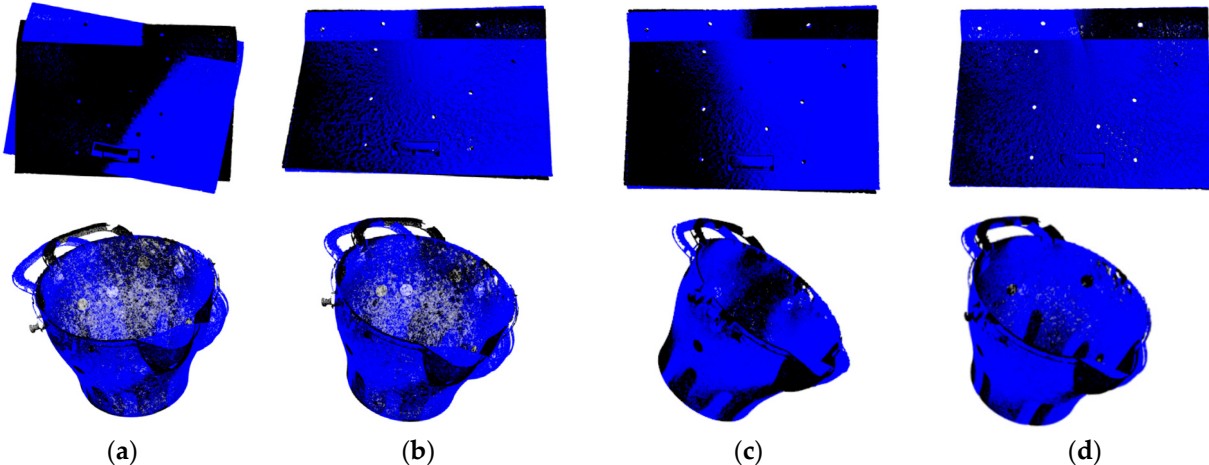

**Figure 8.** Results of workpiece point cloud registration. (**a**) Standard ICP algorithm; (**b**) Standard RANSAC algorithm; (**c**) Depth Filtering-ICP; (**d**) IWOA-RANSAC-ICP algorithm.

The registration time of the point clouds of Workpieces A and B using the three algorithms are shown in Table 3 below.

**Table 3.** Registration time of surface workpiece point cloud (ms).

| Point Cloud | ICP | RANSAC | Depth Filtering-ICP | Algorithm of This Paper |
|---|---|---|---|---|
| A | 961,357 | 13,252,610 | 2,512,542 | 989,790 |
| B | 150,330 | 261,587 | 452,535 | 368,283 |

As can be seen from the above table, when the point cloud data increases significantly, the time of all the above four algorithms increases accordingly. For workpieces A and B, the time difference between the time required by the ICP algorithm and the time of the method designed in this paper is not large. Although the algorithm process in this paper is longer, it still maintains a good alignment speed and accuracy due to the better robustness of IWOA, and the RANSAC and Depth Filtering-ICP algorithms require a larger increase in time and a longer alignment time, which proves that the algorithms need to be further optimized. The mean square error of the point clouds of Workpieces A and B using the four algorithms is presented in Table 4 below.

**Table 4.** Comparison of mean square error of the 4 algorithms.

| Point Cloud | Error | $\sigma_{MSE}$ (mm) | | | |
|---|---|---|---|---|---|
| | | ICP | RANSAC | Depth Filtering-ICP | Algorithm of This Paper |
| A | x | 2.49386 | 2.38141 | 3.245254 | 1.145785 |
| | y | 0.885543 | 0.507815 | 1.251014 | 0.751423 |
| | z | 0.307292 | 0.65359 | 1.321525 | 0.895243 |
| B | x | 1.215468 | 0.856487 | 1.321587 | 0.786325 |
| | y | 0.569874 | 1.325469 | 0.514524 | 0.684751 |
| | z | 0.965124 | 0.956787 | 1.025410 | 0.884265 |

As can be seen from the above Figure 8, Tables 3 and 4, the standard ICP can roughly accomplish the registration of some point clouds with the same downsampling rate and maximum number of iterations throughout the test but is subject to poor registration accuracy, which may be attributed to the higher initial point-to-point requirements of ICP registration and the lower accuracy of subsequent registration in the case of a large initial point pair error. The standard RANSAC algorithm can obtain better results than the standard ICP algorithm in the registration of workpieces A and B. Given that the RANSAC algorithm carries out the registration first based on the fast point feature histogram, it is superior to the ICP algorithm for the registration of workpieces with obvious surface features, but the speed is slower due to the necessity of extracting the histogram first. For Depth Filtering, the ICP algorithm has a significant increase in error and a significant decrease in alignment speed at higher data volumes. The hereby proposed IWOA-RANSAC-ICP algorithm realizes rapid optimization of the RANSAC algorithm using WOA, accelerates the feature extraction, and provides a more accurate initial point pair for the ICP algorithm. Overall, this very algorithm possesses higher accuracy, but its efficiency still needs further improvement.

## 5. Conclusions and Future Work

In this paper, a hybrid optimization algorithm based on the Improved Whale Optimization Algorithm and improved ICP was proposed to achieve point cloud registration in response to the problems and shortcomings of existing point cloud registration methods. In the initial registration stage of this paper, an improved whale optimization algorithm based on nonlinear convergence factor and adaptive weight coefficients was proposed because the number of point clouds is usually substantial, and the standard whale optimization

algorithm is easy to suffer premature convergence that causes it to trap in local optimum as the search proceeds. The algorithm uses smaller weights to search in the neighborhood of the optimal solution in a spiral manner to prevent falling into the local optimum. In the second step of accurate registration, since the standard ICP algorithm is sensitive to noise and outliers, it is prone to misregistration. We proposed an improved ICP algorithm with weighted constraints on the normal vectors of key points to avoid the algorithm from falling into local optimum and further improve the performance of the ICP algorithm in point cloud accurate registration. The experimental results show that the proposed algorithm has lower error compared with other registration algorithms, higher registration accuracy, and faster convergence speed and can meet the needs of subsequent work.

With the increasing accuracy and resolution of visual systems, the obtained point cloud data are becoming more and more informative, and further research is still needed for point cloud denoising strategies while keeping the main cloud point data features and information intact. For the registration of a large amount of point cloud data, there are still limitations in the existing algorithms, and further research is needed to improve the accuracy and efficiency of the registration algorithm.

**Author Contributions:** Formal analysis, L.M.; Funding acquisition, L.W.; Investigation, Y.C.; Methodology, G.L. All authors have read and agreed to the published version of the manuscript.

**Funding:** This paper is supported by the 2021 Jiangsu Higher Education Teaching Reform Research Key Project (No. 2021JSJG156) and Shaanxi Key Laboratory of Machinery Manufacturing Equipment Construction Project.

**Informed Consent Statement:** Written informed consent has been obtained from the patient(s) to publish this paper.

**Conflicts of Interest:** The authors declare no conflict of interest.

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
