# Peer review of "Automatic Registration Algorithm for the Point Clouds Based on the Optimized RANSAC and IWOA Algorithms for Robotic Manufacturing"

_applsci, doi:10.3390/app12199461_

Round 1
Reviewer 1 Report
The authors use the stochastic optimization algorithm IWOA to improve the process of point cloud registration using a binocular stereo camera.
My major objections and remarks:
1. Please provide the view of the test stand. How were placed the binocular stereo camera and portable three-dimensional scanner?
2. Please explain how accuracy was evaluated and provide units to tables 2 and 4.
3. Please add a discussion about computation time; it may be an online method; how many femes per second we may preprocess?
4. A comparison of the local method (ICP) and global optimization method (e.g. WOA) may be unfair. Please add some information on how the initial state was chosen. Also, additional discussion is needed because ICT here works ugly, but I didn't observe it in my research.
5. Please, provide the main novelty presented in the paper because using some optimization algorithm to solve an optimization problem is not the novelty. I see only an additional new experiment for registration with some effectiveness.
There are many mistakes in mathematical notations, e.g.:
1. What symbols: |.| (e.q. 2) and ||.|| (e.q. 14) means? Is it the norm or absolute value?
2. E.q. 13 we have: l_i = sum_i ... - how it depends on i ???
3. E.q. 14 is confusing f(t,i) = ...sum_1 (H(l_i)) how it depends on i ??? also the second equality is incomprehensible.
4. What the black dot (e.q.2, 6) means? Is it an inner product?
Conclusion: Major corrections are needed for acceptance.
Reviewer 2 Report
This paper proposes the combined use of two well known algorithms to analise point clouds for robotic manufacturing. The object is to utilize the advantage of each to improve overall performance. This is demonstrated by using the combined algorithms, RANSAC for initial registration and IICP for refining and producing the final result, against two standard point clouds and several of the authors' own. The results show some useful improvements that need to be reported.
There are a number of minor points that could do with further explanation etc.
line 175-6 This paper concludes in Section 5 is very abrupt. Something like Section 5 discusses the conclusions and provides pointers to further work.
What are European coordinates? I assume you mean Cartesian Coordinates
Line 204 -Should this equation be numbered?
Space after line 280 needs to be removed
Line 317 Figure 2 is very small and needs to ne made bigger
Excess space after line 346
Excess space after line 395
Can tables 1 &2 be split? they would then be clearer
I am not sure what the first line of images in Figure 8 represents. This is not helped by a page break between the first and second row.
Reviewer 3 Report
Could you please let we know:
1- How do you chose convergence factor as the equation 10 and 11?
2- Why the fitness function is used as the equation 14?
meanwhile the notation of the formula need to revised.
Reviewer 4 Report
The paper describes a novel algorithm for point cloud registration that overcomes the limitations of other approaches. The goal is clear and the mathematical formulation is solid and well written.
I have a couple of questions:
The performances of the proposed algorithm are evaluated by providing the complete 3D point cloud of the object.
However, in the real world (and especially in the manufacturing environment mentioned by the title of the paper), the object is typically observed from a single point of view, resulting in an incomplete point cloud, due to self-occlusion. In this case, the observed object has a number of points much smaller than the reference 3d model of the object. Can the authors provide some comments about the expected performances/limitations of the proposed algorithm in this scenario?
Line 20 "the algorithm has [...] high convergence speed" (also in line 174, 389 etc). This statement is not supported by data. Fig. 5 shows the plot just for the IWOA algorithm (without a comparison with the other approaches). Table 1 compares the execution time spent by different algorithms, however, it seems that IWOA is the slowest among them.
Minor comments:
Line 352. The specs of the machine on which the algorithm runs should be provided with more details. What is "a display card of 1060 6G"? Is it an Nvidia GTX 1060? Does the algorithm exploit some acceleration/parallelization provided by Nvidia GPU cores?
Section 4.1 This section should be revised and the English should be improved. For example: "Express the feature" (L367), "Establish a histogram" (L375) etc. Either a subject is missing in these sentences or the whole paragraph should be reformulated using a bullet list.
L377-379 This sentence seems to be placed in the wrong section.
Round 2
Reviewer 1 Report
1. I accept most Authors' answers, but: Please revise mathematical formulas because I still see mistakes. E.g. You still use the same subscript "i" in the two mentioned formulas incorrectly. E.q. line 293, you have l_i= sum_i^n ...., the right side of the equation does not depend on "i" it is only the iterator.
2. I think you should add provided pictures with a view of the experiment.
